# Adaptive Immune Responses, Immune Escape and Immune-Mediated Pathogenesis during HDV Infection

**DOI:** 10.3390/v14020198

**Published:** 2022-01-20

**Authors:** Valerie Oberhardt, Maike Hofmann, Robert Thimme, Christoph Neumann-Haefelin

**Affiliations:** 1Department of Medicine II (Gastroenterology, Hepatology, Endocrinology and Infectious Diseases), Freiburg University Medical Center, Faculty of Medicine, University of Freiburg, 79110 Freiburg, Germany; valerie.oberhardt@uniklinik-freiburg.de (V.O.); maike.hofmann@uniklinik-freiburg.de (M.H.); robert.thimme@uniklinik-freiburg.de (R.T.); 2Faculty of Biology, University of Freiburg, 79104 Freiburg, Germany

**Keywords:** hepatitis D virus (HDV), viral escape, CD8+ T cells, CD4+ T cells, T-cell exhaustion, immune-mediated pathogenesis

## Abstract

The hepatitis delta virus (HDV) is the smallest known human virus, yet it causes great harm to patients co-infected with hepatitis B virus (HBV). As a satellite virus of HBV, HDV requires the surface antigen of HBV (HBsAg) for sufficient viral packaging and spread. The special circumstance of co-infection, albeit only one partner depends on the other, raises many virological, immunological, and pathophysiological questions. In the last years, breakthroughs were made in understanding the adaptive immune response, in particular, virus-specific CD4+ and CD8+ T cells, in self-limited versus persistent HBV/HDV co-infection. Indeed, the mechanisms of CD8+ T cell failure in persistent HBV/HDV co-infection include viral escape and T cell exhaustion, and mimic those in other persistent human viral infections, such as hepatitis C virus (HCV), human immunodeficiency virus (HIV), and HBV mono-infection. However, compared to these larger viruses, the small HDV has perfectly adapted to evade recognition by CD8+ T cells restricted by common human leukocyte antigen (HLA) class I alleles. Furthermore, accelerated progression towards liver cirrhosis in persistent HBV/HDV co-infection was attributed to an increased immune-mediated pathology, either caused by innate pathways initiated by the interferon (IFN) system or triggered by misguided and dysfunctional T cells. These new insights into HDV-specific adaptive immunity will be discussed in this review and put into context with known well-described aspects in HBV, HCV, and HIV infections.

## 1. Introduction

The hepatitis delta virus (HDV) is the smallest known human virus and is a satellite virus of hepatitis B virus (HBV). It was first discovered in 1977 in patients with severe HBV infection [1]. Since then, it has been ascertained that HBV/HDV co-infection causes the most severe form of viral hepatitis [2]. The RNA genome is single-stranded and circular, with a high guanosine/cytidine content, triggering back-folding through intramolecular base-pairing and the formation of a rod-like structure [3,4]. Together with two isoforms of the hepatitis delta antigen (HDAg), which wrap the RNA genome, the ribonucleoprotein (RNP) complex is formed [5]. The large (L)-HDAg (27 kDa) is generated through RNA editing at the amber stop codon to code for tryptophan, causing a protein extension of 19 additional amino acids in comparison to the small (S)-HDAg (24 kDa) [6,7]. RNA editing occurs later during the HDV life cycle, balancing between viral RNA replication and virion assembly, whereas the latter is triggered through the L-HDAg [8,9,10]. The human DNA-dependent RNA polymerase II (DdRP) performs HDV genome replication in a rolling-circle manner, and thus genome replication is independent of the helper virus [11,12]. However, for virion assembly, particle release, and cell entry, the RNP requires the envelope of HBV, consisting of the three hepatitis B surface antigen (HBsAg) isoforms (small, S; medium, M; large, L) [13,14,15]. HBV and HDV enter the host cell via their receptor, sodium taurocholate co-transporting polypeptide (NTCP) [16]. Recent studies indicate that surface glycoproteins from HBV-unrelated viruses, such as hepatitis C virus (HCV), could mediate egress of HDV particles, and thus act as a helper virus for HDV transmission [17]. However, screening of HCV-positive donors for anti-HDV antibodies and HDV RNA found no evidence of HDV infection in HCV mono-infected individuals, suggesting that HCV cannot promote HDV transmission in humans [18,19]. In addition to the well-characterized extracellular spread of HDV, cumulating evidence suggest that HDV also spreads through cell division in an HBV-independent manner [20]. Yet, the clinical relevance of cell-division-mediated spread remains to be determined.

It had been estimated that about 10% of the HBsAg-positive carriers are also HDAg-positive [21]. Recently, meta-analyses have evaluated the number of patients with chronic HDV infection more precisely, with numbers ranging from 12 million to 42 or even 72 million people worldwide [22,23,24]. HDV-infected individuals may be carriers of one out of eight different genotypes (1–8), mainly determined by the geographical region where the infection event occurred, except for genotype 1, which is distributed globally [25]. HDV genotypes differ in their genome sequence by 19–40% and most of them can be further subdivided into two to four sub-genotypes (a–d), characterized by an intersub-genotype similarity of >84–90% across the entire genome [26]. HBV/HDV co-infection emerges either as a simultaneous infection or as a superinfection of a previous HBsAg-positive carrier. Simultaneous infection can lead to fulminant hepatitis or persistent infection; however, 95% of patients resolve the infection [27]. In contrast, spontaneous clearance in superinfection is rare, leading to viral persistence in 90% of cases [28,29]. Chronic HDV infection usually leads to an accelerated progression to cirrhosis, an increased risk of liver decompensation, and hepatocellular carcinoma in comparison to HBV mono-infection [30,31]. Until 2020, the only treatment option for HBV/HDV co-infection was the off-label use of pegylated interferon (pegIFN)-α. However, sustained virological response was merely achieved in only 25% of patients [32,33]. Nucleos(t)ide analogues, such as tenofovir and entecavir, are not effective against HDV, since they only target HBV DNA replication, but not HBsAg production. In 2020, bulevirtide (BLV, formerly Myrcludex B) was approved in the European Union, being the first therapy licensed to treat chronic HDV infection. BLV blocks viral cell entry via NTCP by mimicking the preS1 receptor binding domain of HBsAg. BLV leads to a reduction in HDV RNA levels and liver enzymes in the majority of patients; however, treatment over several years may be necessary for viral clearance, stopping rules still need to be defined, and treatment costs are extremely high.

Although there are additional novel treatment options for chronic HBV/HDV co-infection on the horizon (reviewed recently in [34,35]), it is crucial to understand the interaction of these viruses with the immune system in order to design and monitor future treatment strategies. A specific focus of this review is the mechanisms that lead to the failure of the immune system to clear HDV infection. Furthermore, we will discuss the mechanisms of immune-mediated liver damage.

## 2. Innate Immunity

Unlike HBV, HDV is not invisible (“stealth”) to the innate immune system [36,37,38]. Different mouse models, in addition to cell culture systems and primary hepatocytes (PPH), have demonstrated that HDV interacts with melanoma differentiation antigen 5 (MDA-5), resulting in the production of interferon (IFN)-stimulated genes (ISGs) [39,40,41,42,43]. The exact mechanisms by which MDA-5, the cytosolic RNA sensor of unusually structured RNA, recognizes HDV RNA is not yet understood. Indeed, HDV replication and RNP unpacking occurs in the nucleus, and thus HDV RNA should be shielded from MDA-5 detection [43]. Nevertheless, it was demonstrated that HDV infection in vitro and in vivo leads to an upregulation of type I (IFN-β) and type III (IFN-λ) IFN, and their production was reduced if downstream signaling molecules of the RIG-I-like receptors were inactivated [42]. Although cellular secreted IFN-β and IFN-λ initiated transcription of ISGs in vitro, HDV replication appeared to be resistant to the antiviral attack [43]. Of note, supernatant from HDV-infected cells was sufficient to inhibit hepatitis C virus (HCV) replication in an HCV luciferase reporter system [43]. Thus, HDV might have adapted to the IFN-activated state in the liver by blocking or escaping the IFN system (reviewed in [44]). It even has been suggested that some innate pathways support HDV replication productivity [40], since the IFN system induces RNA-editing enzyme adenosine deaminase (ADAR), which is required for RNA editing to generate L-HDAg, and hence enables viral morphogenesis [45]. It is important to note that MDA-5-mediated production of IFN and ISG induction also contributes to a cytokine milieu that recruits professional antigen-presenting cells, and thus enables priming of functional T cells [46].

## 3. Humoral Immunity

The humoral immune response appears to play a minor role in HBV/HDV co-infection control, since antibodies against HDAg are presumably not neutralizing [47,48]. The antibody response against HDAg appears to have a similar pattern compared to the antibody response against the HBV core antigen (HBcAg) [49]. Upon HDV infection (irrespective of simultaneous or super-infection), antibodies against the S- and L-HDAg are detectable in the plasma in the first month after infection [50,51]. Interestingly, no complete class switch from IgM to IgG antibodies appears; instead, plasma cells produce a mixture of IgM and IgG anti-HDAg antibodies [49]. Both antibody types can be detected during acute and chronic HDV infection and their levels decrease gradually after viral clearance [49,51]. Therefore, anti-HDV antibodies serve rather as a diagnostic tool for a past or current contact of a patient with the virus, rather than a marker for the course of the disease [52]. In the past, before routine pan-genomic HDV-PCR analysis was available, anti-IgM levels served as a marker for disease activity, as association of IgM levels with the degree of histological inflammation was observed [53,54,55]. Of note, although evidence of direct antiviral functions of antibodies is missing, a study from 1990 showed that 41% of the L-HDAg amino acids were part of an immunogenic epitope when analyzed with serum from patients with chronic HDV infection [56]. Hence, since only a minority of studies have examined the diversity, function, and dynamics of the humoral immune response against HDV, future studies using up-to-date methodology should clarify the role of B cells and antibodies in HDV infection.

## 4. Cellular Adaptive Immunity

While a lot was already known about HDV molecular virology in the early 2000s, only a few studies had assessed the immune response in HDV infection [57]. Still, it had been assumed that the cellular adaptive immune response, consisting of CD4+ and CD8+ T cells, would be necessary for HDV clearance, similar to what was suggested by animal models, as well as data from other hepatotropic virus infections, e.g., HCV and HBV [58,59]. In addition to that, the general notion that anti-HDV antibodies are not protective, yet about 90% of individuals with simultaneous HBV/HDV infection clear the infection spontaneously, indicated that T cells play a critical role in this process [27,60]. Most of the insights gained on immune responses in HDV infection were from animal vaccination models, in particular, the woodchuck model. In woodchucks, infectious HDV virions are produced if the animals are co-infected with the woodchuck hepatitis virus (WHV), which then serves as a helper virus for particle packing [61,62,63,64]. Although DNA vaccination against HDV induced an antibody response and HDV-specific T cells, identified through stimulation with HDAg-derived peptides that induced proliferation, HDV infection could not be prevented [65]. Additionally, vaccination studies in mice with DNA vaccines induced CD4+ and CD8+ T cell responses [66,67,68]. Despite the induction of a cellular immune response in these animals, this seemed not to be a correlate of protection against HDV challenge. Thus, the immunogenic role of HDAg and its related cellular and humoral immunity has been debated controversially.

Preliminary results in patient cohorts suggested that, in chronic HDV infection, no T cell response is elicited. In a first pilot study by Nisini et al., virus-specific T cell responses were solely detectable in inactive HDV patients, determined by negative anti-HDV IgM status [69]. This conclusion was further confirmed in 2004, when Huang et al. detected HDV-specific CD8+ T cells only in patients who were HDV RNA negative, and thus had cleared HDV infection [70]. These data indicated that HDV-specific CD4+ and CD8+ T cells occur only in resolving, but not persistent, HDV infection. However, this conclusion was challenged in the last 10 years through the investigation of larger patient cohorts and by applying novel technologies for analysis of virus-specific CD4+ and CD8+ T cells. In the first of these studies, Grabowski et al. analyzed cytokine production in the supernatant of HDAg-stimulated peripheral blood mononuclear cells (PBMC) from patients with chronic HDV infection [71]. By applying this strategy, in 16 out of 17 donors, an HDV-specific cytokine response was detected, including IL-2, IP-10, and IFN-γ secretion, with IFN-γ being the strongest cytokine response detected in most of the donors. Accordingly, it was shown that, in chronic HDV infection, a cellular HDV-specific immune response is present, and these cells are at least, in part, functional. Albeit no direct link between disease parameters was observed, the authors detected that patients who did not respond at all or only to one peptide pool had a tendency towards higher viral loads, suggesting that cellular immunity contributes to the control of HDV infection [71]. More recently, Landahl et al. analyzed HDV-RNA-positive and -negative individuals for CD4+ and CD8+ T cells in response to overlapping peptide (OLP) stimulation [72]. Overall, 53% of individuals had at least one single-peptide CD4+ or CD8+ T cell response, ranging from 0 to 5 peptides. Similar to the study by Gabrowski et al., there was no correlation of clinical parameters with the strength and breadth of the HDV-specicfic T cell response [72]. Moreover, using highly sensitive tetramer-based methods, Keflakes et al. and Karimzadeh et al. were able to detect HDV-specific CD8+ T cells ex vivo in chronically infected patients [73,74]. Taken together, recent studies showed that, indeed, HDV-specific CD4+ and CD8+ T cells are also present during chronic HDV infection; however, it appears that the detection rate of HDV-specific T cells is rather low. In the following, the epitope repertoire, the differential role of CD4+ and CD8+ T cells, and their functional characteristics are discussed in more detail.

### 4.1. CD4+ T Cell Response

The HDV-specific CD4+ T cell epitope repertoire was, so far, only investigated in two studies using OLP covering HDAg, leading to the identification of 18 different HDAg regions targeted by CD4+ T cells [69,72]. The entire L-HDAg was immunogenic, albeit with an accumulation of CD4+ T cell epitopes in the N-terminal region (for a complete list of fine-mapped epitopes, see review [75]). In both studies, CD4+ T cell responses were weak and only detectable after antigen-specific culture for 6 or 12 days, except for one patient with acute HDV infection who displayed ex vivo detectable CD4+ T cells with strong cytokine production [69,72]. CD4+ T cell responses in this acute super-infected patient indicate that an initial strong response decreases over time and the CD4+ T cell epitope repertoire narrows during progression to chronic HDV infection [72]. In this study, in silico prediction as well as human leukocyte antigen (HLA)-binding assays were performed to define the HLA restriction of the CD4+ T cell epitopes, resulting in the identification of 14 HLA class II restricted epitopes [72]. Of note, 3 of these 14 epitopes were targeted in more than one patient. Strikingly, eight patients (four HDV RNA positive and four HDV RNA negative) recognized HDAg_41–60_; however, these patients expressed a diverse set of HLA class II alleles. In line with this observation, in silico prediction revealed that 12 HLA class II molecules display a high (half-maximal inhibitory concertation (IC50) < 1000 nM) or intermediate (IC50 1000–5000 nM) binding affinity. In in vitro binding experiments, the epitope could be pinned down to four major histocompatibility complex (MHC) class II molecules (DRB1*08:02/10:01/11:01//15:02). Notably, Nisini et al. also detected a T cell response against this HDAg region in three patients with inactive disease and divergent HLA class II alleles. This promiscuous restriction is not unusual for CD4+ T cell epitopes, since the binding pocket of HLA class II molecules has an open pocket and, therefore, tolerates variable peptides [76,77]. These data indicate that HDAg_41–60_ is an immunodominant CD4+ T cell epitope due to its promiscuous HLA class II binding. However, the HDV-specific CD4+ T cell epitope repertoire has only been analyzed in two studies; thus, larger cohorts should be analyzed in future studies and the remaining CD4+ T cell epitopes need to be fine mapped.

CD4+ T cells have many different functions during viral infections, but they are best known for providing help to CD8+ T cells and B cells, as well as for recruitment of immune cells via the secretion of cytokines. Interestingly, unlike in other hepatotropic viral infections, HBV/HDV co-infection showed the highest percentage of cytotoxic CD4+ T cells in a comparative study of HBV, HCV, and HBV/HDV mono/co-infected individuals; however, CD4+ T cells were studied on an antigen-nonspecific level [78]. The role of cytotoxic CD4+ T cells has been best described in human immunodeficiency virus (HIV) infection [79]. These cells have a high degree of perforin expression and share common features with cytotoxic CD8+ T cells; thus, they are able to kill virus-infected cells [79]. In chronic HDV infection, these CD4+ T cells displayed the phenotype of terminally differentiated effector cells, indicated by loss of the co-stimulatory molecules CD28 and CD27. In addition, their numbers were increased in patients with advanced liver disease [78]. A previous study had already found that some CD4+ T cells showed cytotoxic potential in specific assays [69]. This phenotype was attributed to either the T helper-1 (TH1) or Th0 CD4+ T cell subset, as these cells display a cytotoxic activity and produced high amounts of IFNγ, a hallmark molecule of both subsets (Figure 1A) [69,80,81]. The phenotypic characterization of single-epitope-specific CD4+ T cells in other hepatic viral infections, such as HBV and HCV, has elucidated the role of CD4+ T cells in infection control [82,83,84]. So far, nothing is known about the functional competence of CD4+ T cells in chronic HDV infection and whether these cells might display markers of dysfunction or exhaustion. Although CD4+ T cell epitopes have been fine-mapped and the tools for peptide-loaded HLA class II tetramers are available, investigations of single-epitope specific CD4+ T cells in HBV/HDV co-infection are still lacking. Thus, the impact of chronic antigen stimulation on the HDV-specific CD4+ T cell compartment is still unclear. Especially in light of the presence of high amounts of cytotoxic CD4+ T cells, high-dimensional single-epitope analysis should be performed to define their role in chronic HDV infection in comparison to other hepatitis virus infections. CD4+ T cells have a predominantly TH1 phenotype, targeting epitopes mainly in the N-terminal region of HDAg, which show a promiscuous HLA class II restriction; however, the role of CD4+ T cells in HDV infection control is still only poorly understood. Studies assessing the fate and function of CD4+ T cells during antiviral therapy might contribute to clarifying the impact of CD4+ T cells on infection outcome.

### 4.2. CD8+ T Cell Response

Early studies in woodchucks showed that immunized animals had a reduced level of HDV viremia in the absence of detectable circulating anti-HDV antibodies, indicating that cytotoxic T cells contributed to viral control [85]. Since these experiments were performed in woodchucks that have a different MHC class I repertoire than humans, the epitopes presented are different and, therefore, these results cannot simply be applied to humans. In order to avoid this limitation, the vaccination model was transferred to mice with expression of a human HLA class I molecule (HLA-A*02:01) [70]. Huang et al. predicted potential HLA-A*02:01-binding peptides in silico and used promising candidates for HLA-A*02 tetramer generation. In transgenic mice infected with an HDV DNA vaccine, 0.9% of the splenic CD8+ T cells were either HDAg_26–34_ or HDAg_43–51_. The results obtained in the mouse model were verified in patients who were anti-HDV-positive; notably, only in patients with negative HDV RNA were HDAg_26–34_ and HDAg_43–51_-specific CD8+ T cells detected after PBMC expansion [70].

For the next 14 years, these two HLA-A*02:01-restricted epitopes were the only CD8+ T cell epitopes known for HDV. However, in the last years, additional studies were conducted to broaden the HDV-specific CD8+ T cell epitope repertoire. Two studies used in vitro expansion of PBMC with OLP libraries of HDAg [72,73]; two international collaborative studies determined HLA class I-associated viral sequence polymorphisms, followed by in silico peptide binding prediction to identify novel HDV-specific CD8+ T cell epitopes [74,86]. Indeed, Karimzadeh et al. first screened for HDV peptides with good binding to frequent HLA class I alleles (in Europe, HLA-A*01, A*02, -A*03, A*24, B*07) or the HLA allele B*27 that has a dominant role in restricting protective CD8+ T cell responses, e.g., in HIV and HCV infection [87,88,89,90]. Strikingly, only two peptides, restricted by HLA-B*27 (HDAg_99–108_ and HDAg_104–112_), were confirmed in three patients with resolved HDV infection but in none of the five patients with chronic HDV infection [86]. In another study by Karimzadeh et al., HDV viral sequences from 104 patients with chronic HDV infection were analyzed for HLA-associated sequence polymorphism (HLA footprints) [74]. HLA footprints indicate that virus-specific CD8+ T cells restricted by the respective HLA class I allele target the viral region flanking this HLA footprint, leading to viral evolution that enables viral escape from the CD8+ T cell response (viral escape will be discussed in more detail below). Overall, 21 HLA-associated polymorphisms were detected; for five of these, corresponding CD8+ T cell epitopes were verified experimentally. Interestingly, most of these epitopes were restricted by rare HLA class I alleles. The most striking example in this study was the HLA-B*15:01-restricted CD8+ T cell epitope HDAg_170–179_; 10 out of 14 HLA-B*15:01-positive patients with resolved or persistent HDV infection displayed a T cell response against this epitope [74]. In sum, these two studies point towards an HDV-specific CD8+ T cell epitope repertoire that is dominated by rare HLA class I alleles.

Next to these viral sequence-based approaches, Landahl et al. and Kefalakes et al. used an unbiased strategy, leading to the identification of novel HDV-specific T cell epitopes, with some overlapping the epitopes identified by the sequence-based approach [72,73]. Although Landahl et al. used a library of OLP with a length of 20 amino acids, they identified CD8+ T cell responses in 35% of patients, ranging from 0 to 4 positive peptides. In silico prediction of HLA class I binding suggested that the majority of responses might be restricted by the HLA-B*07 supertype family (HLA-B*35:01, B*51:01, and B*53:01). Intriguingly, the CD8+ T cell epitopes accumulated in the C-terminal region of the L-HDAg [72]. In line with this was the OLP screening by Kefalakes et al., detecting CD8+ T cell responses in 71% of the patients, mainly mapping to the C-terminal region of HDAg [73]. Four epitopes in the C-terminus overlapped and were restricted by HLA-B*07, B*35:01, B*52:01, and B*58:01. Additionally, an HLA-B*18:01 and B*27:05-specific CD8+ T cell epitope was identified, corresponding to the epitopes discovered by Karimzadeh et al. [74,86]. The detection rate of HDV-specific CD8+ T cells was increased by applying newer technologies of single-epitope specific CD8+ T cell characterization with peptide-loaded (p)MHC tetramer technology. Similar to chronic HBV and HCV infection, the frequency of HDV-specific CD8+ T cells was extremely low [73,74]. In total, 17 different CD8+ T cell epitopes were described. For most of these epitopes, epitope fine-mapping and HLA restriction experiments were performed, while, for a smaller number of epitopes, fine-mapping and HLA restriction were only performed in silico ([70,72,73,74,86] and recently reviewed in [75]). Of note, the vast majority of defined HDV-specific CD8+ T cell epitopes are restricted by HLA-B alleles. So far, almost no data of patients during acute HDV infection are available; thus, information of the CD8+ T cell repertoire and the breadth of response in this decisive phase of infection is still missing. More data on HDV-specific CD8+ T cell responses during acute infection are needed to improve the understanding of spontaneous infection control and guide the way for immunotherapy in chronic infection.

### 4.3. Failure of the T Cell Response in Chronic Infection

Some viral infectious diseases are rapidly controlled by the immune system, resulting in elimination of the virus, whereas other viral infections are not successfully controlled by the immune systems, resulting in chronic infection. Host as well as viral factors determine this differential outcome of infection. In particular, cytotoxic CD8+ T cells detect and eliminate cells with replicating virus. However, if viral infection persists, increasing inflammation and constant antigen exposure hamper CD8+ T cell functionality. In animal models of persistent viral infections, as well as in human chronic infections, such as HIV, HBV, and HCV, two main mechanisms that contribute to the failure of CD8+ T cell response have been described: T cell exhaustion and mutational viral escape (Figure 1).

#### 4.3.1. Viral Escape

In HCV and HIV infection, viral escape has been intensively studied [91,92] and recently reviewed in [93]. Three mechanisms that lead to an abrogated or diminished CD8+ T cell peptide recognition are known: amino acid changes in the region (i) of the binding pocket of HLA class I molecules, (ii) of the T cell receptor (TCR) interaction region [94], and (iii) flanking the epitope, leading to a failure of optimal epitope processing and presentation (Figure 1B) [92,95,96].

In the case of HDV, the exploitation of the human DdRP should protect the virus from sequence variation, since the DdRP has a low error rate due to its proofreading activity [97]. However, for HDV replication, a template switch from DNA to RNA has to occur, potentially causing a higher genetic diversity [98,99]. Furthermore, the S-HDAg binds the DdRP, thereby accelerating the translocation of the polymerase on the cost of fidelity, triggering sloppy template recognition and nucleotide integration [100]. New data using next generation sequencing have verified earlier results [101] that HDV has a high complexity, similar or higher than that of other RNA viruses with a mutation rate of 9.5 × 10^−3^–1.2 × 10^−3^ substitutions/site/year [101,102,103,104,105]. Analyses of nucleotide polymorphisms in five patients with chronic HDV infection revealed that these polymorphisms were associated with a positive selection, since they occurred in potential immunological epitope regions [106]. First experimental evidence for viral escape in HDV infection was obtained for the two HLA-B*27-restricted CD8+ T cell epitopes that are located in a relatively conserved viral region (HDAg_99–108_ and HDAg_103–112_). Comparative sequence analyses of HLAB*27-positive and -negative patients revealed that two amino acid substitutions were significantly enriched in HLA-B*27-positive patients [86]. This HLA footprint proved to be functionally relevant in in vitro stimulation experiments and with molecular modeling, revealing that the structural and electrostatic properties of the bound peptides differed considerably at the TCR interface. The HDV sequence database generated in this study was used to define HLA footprints for all HLA class I alleles, not only HLA-B*27, leading to the identification of novel epitopes [74]. It was demonstrated that HDV variant epitopes were only partially recognized by CD8+ T cells isolated from HDV-infected patients, indicating that the virus had escaped detection by these cells. In the case of an HLA-B*18-restricted CD8+ T cell epitope, only the variant peptide and not the consensus sequence triggered an IFNγ release. Of note, the consensus sequence with amino acid D47 was enriched in HLA-B*18-positive patients, while the “variant” E47 was more frequent in HLA-B*18-negative patients. These data indicate that the E47 variant was the previous prototype sequence; however, viral escape from the HLA-B*18-restricted CD8+ T cell response led to the accumulation of the D47 sequence. Most likely, this substitution does not impair viral fitness, leading to the fixation of this sequence also upon transmission to an HLA-B*18-negative patient. Thus, viral escape mutations are a driver of viral evolution on a population level [74]. This is in line with another observation of this study: HDV-specific CD8+ T cell epitopes were restricted only by rare HLA class I alleles. These combined findings indicate that HDV has adapted well on the population level to avoid immune recognition in the context of common HLA class I alleles.

Of note, Kefalakes et al. observed viral escape in 6/11 (54%) analyzed CD8+ T cell epitopes targeted in the respective patient [73]. It was observed that the mutation rate is not uniform within the HDV genome [107] and during the course of infection [105]. Homs et al. illustrated that, in chronic HDV infection, the mutation rate decays over the time elapsed, indicating that HDAg initially displays a high mutation rate to adapt to the new host and then reaches a “steady state” [105], similar to HCV infection [108,109]. The early viral adaption to the host HLA class I alleles was shown by Karimzadeh et al. in an HDV super-infection event of an HLA-B*15:01-positive patient. Through Sanger sequencing, the L-HDAg changes were analyzed longitudinally [74]. Following 71 weeks after infection, four sequence variations from the initial viral sequence had occurred. Strikingly, one of the substitutions (S170N) was a bone fide escape variant located in HLA-B*15:01-restrcited L-HDAg_170–179_ CD8+ T cell epitope, which was detected in all HDV-infected HLA-B*15:01-positive patients. As this variation is an experimentally proven escape mutation, it is tempting to speculate that its appearance during HDV super-infection could cause a diminished CD8+ T cell response, and thus provokes the failure of viral clearance, leading to HDV persistence. Single-epitope specific CD8+ T cell characterization revealed that the loss of cognate antigen recognition triggered phenotypical changes [73,74], consistent with findings that were made during viral escape in HCV infection [110,111]. Both studies observed that T-cell-targeting escaped epitopes had a diminished expression of the activation marker CD38, intermediate expression of PD-1, which is an activation but also regulatory protein, and a rather high expression of CD127, the receptor of IL-7a that is important for homeostasis [73,74]. Correspondingly, these CD8+ T cells expressed a low level of the transcription factor T-bet and Eomes and a high amount of TCF1, a transcription factor that defines memory-like cells with proliferative potential [112]. This observation is in line with findings in HCV and HIV infection, where the loss of antigen recognition mediated by viral escape led to the formation of a T cell subset that is similar to the memory T cell compartment in acute-resolving infection and that was called memory-like [110,111,113,114]. In sum, viral escape appears to play a major role in the failure of HDV infection control, since the race between viral host adaption and antiviral T cell pressure might determine the outcome of infection.

#### 4.3.2. T Cell Exhaustion

In contrast to other viral infections, T cell exhaustion has so far not been well studied in HBV/HDV co-infection [115]. T cell exhaustion was first defined in the LCMV mouse model [116,117,118], where constant cognate antigen stimulation caused dramatic changes in the CD8+ T cell phenotype, including metabolic, epigenetic, and transcriptomic changes [119,120,121,122]. Exhaustion was described as a sequential and hierarchical process from cytotoxic polyfunctional T cells towards a decreasing effector potential and sustained upregulation and co-expression of different inhibitory receptors [123]. This includes the loss of versatile cytokine production and proliferation, impaired cytotoxicity and the expression of inhibitory receptors, in particular, PD-1, CD160, 2B4, Tim-3, and CD39, as well as the upregulation of specific transcription factors, such as TOX [124,125,126,127,128]. In chronic HCV infection, virus-specific CD8+ T cells that target conserved (non-escaped) epitopes display a rather exhausted phenotype, while CD8+ T cells targeting escaped epitopes display a more memory-like phenotype [111,129]. The role of T cell exhaustion in chronic HBV infection is a current research focus and might depend on the targeted antigen [126,130,131,132].

Since HDV infection displays similar dynamics concerning viral escape compared to HCV infection, it is conceivable that CD8+ T cells targeting epitopes in conserved regions display an exhausted phenotype, and thus T cell exhaustion contributes to CD8+ T cell failure. However, the analysis of CD8+ T cells not affected by viral escape mutations has been difficult, as the majority of detectable CD8+ T cells target epitopes subjected to viral escape. Yet, the studies by Karimzadeh et al. and Kefalakes et al. indicate that T cells directed against the prototype antigen have a more chronically activated phenotype, indicated by the expression of higher levels of CD38, PD-1, and a decreased expression of CD127, as well as lower levels of TCF-1 and the pro-survival factor BCL-2 [73,74]. Moreover, a previous report could rescue the HDV-specific CD8+ T cell response by applying IL-12 rather than blocking of the inhibitory receptors PD-1 and CTLA4 [133]. This indicates that HDV-specific CD8+ T cells might not be efficiently activated or are in a state of exhaustion that cannot be easily restored by checkpoint inhibition. In conclusion, the exact contribution of CD8+ T cell exhaustion to HDV persistence needs to be further investigated, ideally with HDV-specific CD8+ T cell epitopes that are not prone to viral escape in the majority of patients.

### 4.4. Hypothesis of Factors That Influence the HDV-Specific CD8+ T Cell Repertoire

Strikingly, all HDV-specific CD8+ T cell epitopes identified in chronic and resolved patients so far were restricted by HLA-B alleles, except for the two HLA-A*02-restricted epitopes already identified in 2004. There might also be a bias towards HLA-B-restricted epitopes in other viral infections [134,135,136], likely due to the fact that HLA-B alleles are the most polymorphic HLA class I alleles; however, the almost exclusive restriction of HDV epitopes by HLA-B alleles could also point towards a functional relevance. It has been implied from analysis of PBMC in HIV, CMV, and EBV infection that T cells restricted by HLA-B alleles have a decreased T cell receptor (TCR) avidity towards the antigen, together with a phenotypical profile of reduced effector potential in comparison to HLA-A-restricted T-cells [137]. However, these data are conflicting with previous studies that detected a higher functional avidity of HLA-B-restricted responses, which correlated with a superior cytokine profile, rapid target cell lyses, and more efficient eradiation of infection [138,139,140]. T cells with a higher avidity are more susceptible to activation-induced cell death and the higher degree of stimulation could induce T cell exhaustion [141,142]. Additionally, T cells with a higher avidity cause more immunological pressure; thus, epitopes targeted by these cells might be more prone to viral escape [143,144,145]. The HDV genome is highly flexible, tolerating many sequence variations. Thus, it appears that HDV can rapidly adapt to its host to avoid elimination by cytotoxic T cells, which might affect the gene pool of HDV on a population level. Hypothetically, former HLA-A-restricted T cell epitopes, which had a higher TCR avidity, could have caused a stronger immunological pressure than HLA-B allele-restricted T cells, and hence these epitopes underwent viral escape, the variations were fixed, and the epitopes were lost from the repertoire. An alternative explanation might be that viral escape occurs at a similar frequency in HLA-A- and HLA-B-restricted epitopes, but the escape on a population level from HLA-B alleles is more difficult, as these have a larger inter-human diversity concerning allele loci and HLA supertypes [146]. Consequently, immunological pressure on HLA-B-restricted epitopes is often lost after viral transmission, since expression of the corresponding HLA-B allele is infrequent in the subsequent host. However, HLA-A alleles are less diverse, with the majority of the European population being either HLA-A*01, -A*02, or -A*03 positive, thus viral escape in these epitopes is more likely fixed in the population, leading to loss of the respective epitopes.

Additionally, it needs to be considered that HBV and HDV replicate in the same hepatocyte; thus, presented epitopes on the HLA complexes derive from both HBV (core, polymerase, surface, and X) and HDV (HDAg) proteins. Which peptides are presented and their half-life on the HLA complex depend on different factors, including proteasomal processing, TAP transportation, and HLA binding affinity, as well as the quantity of the respective protein available for peptide processing [147,148,149,150,151]. Thus, peptides might have a high HLA affinity, but, if they are not well cleaved from the protein and transported into the endoplasmic reticulum (ER), another peptide will be the predominately presented one. Thus, hepatocyte detection and elimination depend on the presented epitope and the functionality of CD8+ T cells targeting the epitope. A recent study indicated that, in HDV-infected liver samples, transcription of genes that are involved in peptide processing (TAP2) are upregulated in comparison to HBV mono-infection and uninfected livers [152]. Furthermore, the biased epitope presentation might already determine T cell priming and, therefore, influence the CD8+ T cell repertoire. To this end, no studies have investigated and compared the HBV-specific CD8+ T cell epitope repertoire in HBV mono- vs. HBV/HDV co-infection. For HBV and HDV, HLA-A*02:01 and B*35:01 epitopes have been well described [70,73,131]; thus, with the help of in silico tools, the peptide processing and presentation on the HLA class I complexes can be compared (Table 1). These data suggest that, in the case of HLA-A*02:01 epitopes, the HBV-derived core_18–26_ will dominate the pool of presented epitopes, since it has a good processing efficiency combined with a high HLA class I affinity. This might result in a decreased presentation of HDV-derived epitopes and, therefore, HDV-specific CD8+ T cells might not be activated or not even induced. In contrast, in the case of HLA-B*35:01, the in silico prediction would rather indicate a bias towards HDV-derived epitopes (Table 1). Certainly, the amount of protein available for processing might have an additional impact on peptide presentation. It was suggested that, in woodchuck infected livers, up to 6 million copies of the HDAg per cell can be found [153]. Albeit similar data for HBV-infected livers are missing, it is known that the most abundant protein in HBV-infected cells is HBsAg, followed by HBcAg and the polymerase. Hence, HDAg likely competes with HBsAg and HBcAg for peptide processing and presentation. Interestingly, one study observed a correlation of detected HDV-specific T cell response with decreasing HBV viral load [72]. It is tempting to speculate that a decrease in HBV particle production, and thus HBV protein translation, could lead to an increase in HDV-derived epitope presentation and, therefore, increased T cell detection. Nevertheless, a study examining the immunopeptidom of hepatocytes co-infected with HBV and HDV could elucidate the impact of protein quantity, as well as processing of peptides, and thus the impact on distribution of immunodominant CD8+ T cell epitopes in HBV/HDV co-infection.

## 5. Immunopathogenesis in HBV/HDV Co-Infection

HBV/HDV co-infection causes the most severe form of viral hepatitis, leading to an accelerated progression to liver cirrhosis [2]. Liver biopsies revealed that, in HBV/HDV co-infection, the degree of liver damage is almost twice as high as it is in HBV or HCV mono-infection due to the occurrence of severe lobular inflammation and necrosis [160]. Since HDV induces an interferon response, it was suggested that the increased cytokine levels might lead to a higher degree of liver inflammation, resulting in a more severe course of infection in comparison to chronic HBV mono-infection. Furthermore, early experiments in transgenic mice expressing HDAg in their hepatocytes led to the conclusion that HDAg itself is not cytotoxic, since mice did not show signs of liver pathology during the studied period of 18 months [161,162]. Recent analysis of mice infected with adeno-associated vectors (AAV) carrying replication-competent HBV and HDV genomes challenged this initial observation and raised the hypothesis that HDAg itself, in particular, S-HDAg, could contribute to liver damage even in the absence of T cells [163,164]. Usai et al. described this as part of a bimodal mechanism leading to liver inflammation, which involved, as a second mechanism, tumor necrosis factor (TNF)-α. In the same mouse model, they demonstrated that the inhibition of TNFα signaling via the agonist etanercept significantly reduced ALT serum levels, whereas no improvement was observed if infected mice had a gene knockout (KO) for signaling of type I and type II IFN (IFNα/βR KO, IFNγR KO, MAVS KO) [164]. Moreover, Suárez-Amarán et al. observed that *TNF**α* transcription is induced in the liver of AAV-HDV-infected mice independent of the adaptive immune response and likely independent of MAVS signaling [42]. In line with this is the finding that, in Huh7 and HEK293 cells, L-HDAg interferes with the TNFα–NF-κB signaling axis by promoting different steps of the pathway [165]. TNFα is an inflammatory cytokine contributing to liver inflammation and, if enduring, this leads to liver fibrosis and eventually cirrhosis (reviewed in [166]). Strikingly, TNFα correlates with HDV-RNA levels, as well as disease progression in chronic HDV-infected patient serum [167]. Besides TNFα, transforming growth factor (TGF)-β is a major regulator of liver fibrosis and cirrhosis [168]. In cell culture models, it was described that the TGF-β-c-Jun-induced signaling cascade is boosted by L-HDAg [169]. However, TGF-β serum levels were reduced in chronic HBV/HDV co-infected and HBV mono-infected patients in comparison to healthy controls, including also later disease stages [167]. Moreover, L-HDAg was also attributed to induce oxidative stress, which, in turn, also activates NF-κB and STAT-3, leading to a dysregulation of inflammation, apoptosis, and invasion contributing to cirrhosis and HCC formation [170,171]. Altogether, the abovementioned pathways trigger a cellular immune mechanism, including innate and adaptive responses (Figure 2A).

Viral infections require a balance of attacking virus-producing cells and, at the same time, maintaining tissue function. A recent study by Tham et al. highlighted another aspect of HDV modulation of the liver environment [152]. Mediated by the HDV-induced IFNβ and IFNλ signaling, an upregulation of genes involved in antigen processing and presentation in primary human hepatocytes infected with HDV and in neighboring cells was detected. This observation was transferred to patient biopsies, in which an upregulation of genes involved in antigen processing was measured. In vitro experiments then proved that the upregulation of antigen-processing machinery resulted in a higher efficiency of HBV epitope presentation in HBV/HDV co-infected cells in comparison to HBV mono-infected cells, which could promote T cell recognition of infected cells [152]. However, it was suggested that, since HBV-specific CD8+ T cells have a reduced effector function, the upregulation of HBV-derived epitopes does not lead to viral clearance, but rather CD8+ T cell activation, resulting in increased cell infiltration, and thus sustained inflammation (Figure 2B) [152].

Albeit experiments in an AAV-HDV mouse model indicate that leukocytes are recruited to the pathogenic liver, liver function was not restored if T and B cells (Rag1 KO mice), NK cells (α-NK1.1), or macrophages (clodronate-loaded liposomes) were depleted in infected animals [164], suggesting that liver damage in HDV infection is not caused by the cellular arm of immune response. Nevertheless, a recent study analyzed, for the first time, liver-infiltrating immune cells in liver biopsies from 24 chronic HBV/HDV co-infected patients [160]. Thereby, the degree of activation of the innate populations of natural killer (NK) cells and mucosa-associated invariant T (MAIT) cells in the blood and in the liver was studied in addition to the analysis of HDV-specific and bulk CD8+ T cells. The analyses revealed that there is an enrichment of MAIT, NK, and HDV-specific CD8+ T cell in the liver in comparison to the circulation. However, MAIT and NK cells were reduced in the liver and blood in comparison to uninfected controls, similar to previous observations that indicated that MAIT cells are functionally impaired, leading to subsequent loss of MAIT cells in the blood, which could contribute to HDV-associated liver pathology [172]. Correspondingly, liver-resident cells were activated and, most likely, although only indicated by the degranulation-associated molecule CD107a, had a high degree of effector potential expanding also to bystander nonspecific CD8+ T cells [160]. This included an increased expression of the activating receptor NKG2D on MAIT, NK, and CD8+ T cells. Furthermore, the expression of NKG2D of intrahepatic CD8+ T cells was positively correlated with CD107a expression, and thus degranulation that correlated with liver enzyme activity and aspartate aminotransferase-to-platelet ratio index score. The authors, therefore, concluded that there is a general antigen-nonspecific activation of the resident memory CD8+ T cells contributing to disease stage and inflammation [160]. The increased expression of NKG2D on liver-infiltrating CD8+ T cells was also observed in chronic HBV and HCV infection, and likewise correlated in HCV infection with a higher activity of liver enzymes and a greater histological severity of liver injury [173,174]. Whereas NKG2D acts as a direct activator on NK cells, triggering cytotoxicity and cytokine secretion [175,176,177], Kennedy et al. demonstrated that it is a co-stimulatory molecule acting in synergy with the TCR on HBV-specific CD8+ T cells, similar to other viral infections [174,178]. Consequently, the sole expression of NKG2D on CD8+ T cells in the liver would not directly lead to more hepatocyte death, at least in HBV [174]. The question arising is, might there be a difference in NKG2D ligand expression in HBV mono- in comparison to HBV/HDV co-infection, resulting in an increased liver pathology in co-infection? The ligand of NKG2D is MHC class-I-related chain (MIC)-A/B that is induced by cellular stress, viral infection, and IL-15 [179,180]. Conversely, to HBV, HDV is sensed by the innate immune system and induces IFNβ and λ, which could lead to an increased expression of MIC-A/B on hepatocytes, as well as an increased IL-15 secretion in the liver, inducing NKG2D expression on lymphocytes [174,181]. Kefalakes et al. observed in in vitro experiments in an HDV-producing cell line (HuH7-END) that HDV-positive cells compared to HDV-negative cells upregulate MIC-A/B, indicating that HDV-infected hepatocytes might be a target of NKG2D-expressing cytotoxic cells [160]. Unfortunately, they did not compare MIC-A/B levels in an analogous infection system of HBV mono-infected cells. However, similar experiments were conducted in HepG2 cells, indicating that HBV downregulates MIC-A/B expression, potentially through HBsAg, which, in turn, activates miRNAs inhibiting MIC-A/B mRNA translation [182,183]. On the other hand, an increased expression of MIC-B was detected on HCV-infected hepatocytes, although this was not a definite observation [174]. Therefore, a hypothesis could be that, in HBV mono-infection, MIC-A/B is reduced; thus, infiltrating lymphocytes have a reduced activation and, hence, a diminished lysis of infected hepatocytes, whereas additional infection with HDV triggers a surface increase in MIC-A/B, leading to accelerated liver damage (Figure 2C). Nevertheless, a comparative study of HCC tissue samples (including healthy liver sections in the same patient) from HBV-, HCV-, and HDV-infected patients did not state and, hence, observe a significant differential expression of MIC-A/B in distinct viral infections [184]. Consequently, as this is the only comprehensive analysis of patient liver samples, it will be important that future studies will assess the expression level of NKG2D ligands on hepatocytes of hepatitis patients in order to elucidate the role of NKG2D-mediated liver pathology. Altogether, it is evident that increased liver pathology in CHD cannot be attributed to one mechanism but is the sum of many dysregulated pathways that need to be studied in depth, especially in liver samples of patients, to further clarify the picture.

## 6. Conclusions

A lot of effort has been made in the last five years in order to elucidate the mechanism behind HDV-associated severe liver pathology and adaptive as well as innate immune factors contributing to this. Yet, there are still many open questions concerning especially the role of HDV-specific CD8+ and CD4+ T cells. With the recent report of liver samples from CHD patients, the assumption was raised that liver pathology is caused by an antigen-unspecific manner of liver-resident CD8+ T cells [160]. Future studies have to focus on this, together with the aspect of the interplay between HBV- and HDV-specific CD8+ T cells and the potential effects of increased viral protein epitope presentation [152]. Moreover, it will be of great interest how BLV inhibiting HBV and HDV extracellular spread will influence the liver homeostasis and if liver functionality will be repaired. Notably, immunological studies of chronic HBV/HDV co-infected patients were almost exclusively performed in Europe and North America, hence in the background of HDV genotype 1 infection. Consequently, the described epitopes and virus-specific T cells to date were all derived from genotype 1 HDV infections. It will be exciting to see if viral evolution to avoid immune recognition will be also detectable in genotypes that are not globally distributed, but rather endemic in the context of the respective dominant HLA alleles in those regions.

## Figures and Tables

**Figure 1 viruses-14-00198-f001:**
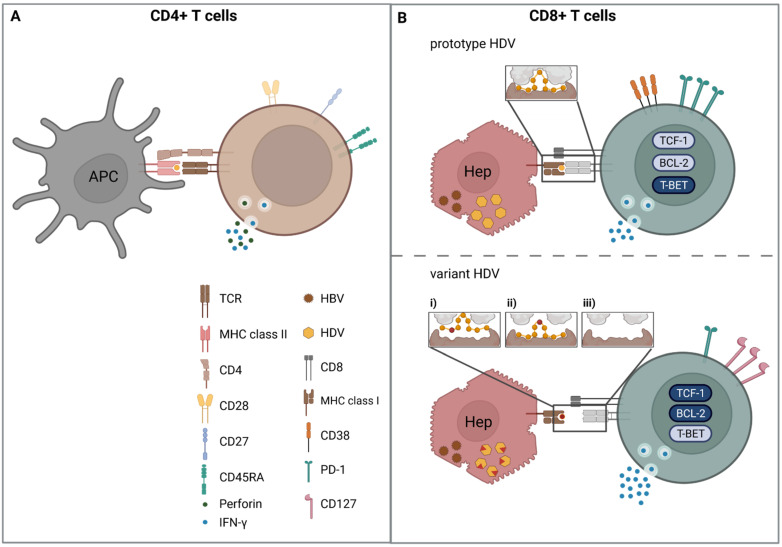
(**A**) CD4+ T cells secrete IFNγ in response to peptide presented by an APC. Some cells are cytotoxic CD4+ T cells that are perforin-positive, have a low expression of CD27 and CD28, and a varying expression of CD45RA. (**B**) Mechanisms of viral escape. Upper panel: the hepatocyte is infected with prototype HDV; thus, the CD8+ T cell detects cognate peptide and has a phenotype of persistent activation. Lower panel: the hepatocyte is infected with variant peptide; therefore, peptide presentation is lost due to (i) AA variation in MHC class I binding anchor position, (ii) AA variation in TCR interaction region, (iii) AA variation in epitope flanking region, resulting in peptide processing failure. As a consequence, the CD8+ T cell does not recognize its antigen anymore and is in a phenotypic memory-like state. Dark blue, high expression; light blue, low expression; APC, antigen-presenting cell. Created with BioRender.com, 1 December 2021.

**Figure 2 viruses-14-00198-f002:**
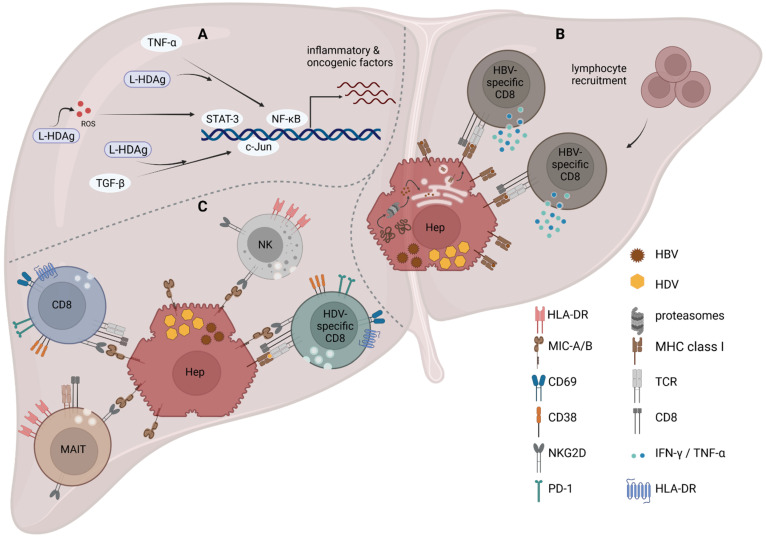
A variety of molecular and cellular pathways contribute to liver inflammation in HBV/HDV co-infection, causing liver cirrhosis. (**A**) L-HDAg can interact with different signaling pathways in a direct or indirect manner, resulting in amplified cytokine responses. This results in an increased activation of the transcription factors STAT-3, c-Jun, and NF-κB and transcription of genes contributing to inflammation. (**B**) HDV induces ISGs, including proteins involved in the antigen presentation pathway. In consequence, compared to HBV mono-infection, more HBV epitopes are presented on the cell surface, resulting in an increased activation of HBV-specific CD8+ T cells and general lymphocyte recruitment. (**C**) Increased numbers of NK and MAIT cells are detected in HBV/HDV co-infected livers, as well as increased numbers of HDV-specific CD8+ T cells. CD8+ T cells have a tissue resident phenotype. MAIT, NK, and CD8+ T cells are in an effector state indicated by degranulation and activation molecule expression, among them NKG2D, which recognizes its ligand MIC-A/B on infected hepatocytes. Created with BioRender.com, 1 December 2021.

**Table 1 viruses-14-00198-t001:** MHCI binding predictions were made on 2 November 2021 using the IEDB analysis resource Consensus tool [154], which combines predictions from ANN aka NetMHC (4.0) [155,156,157], SMM [158], and Comblib [159]. For all values, higher scores indicate higher predicted efficiency, whereas a smaller MHC IC50 predicts a better binding. The processing score combines the proteasomal cleavage and TAP transport predictions. The total score combines the proteasomal cleavage, TAP transport, and MHC binding predictions. The sequences highlighted in grey show the best candidate for the indicated MHC class I complex.

Protein	AA Position	Sequence	HLA-Allele	Proteasome Score	TAP Score	MHC Score	Processing Score	Total Score	MHC IC50 [nM]
HBcAg	18	FLPSDFFPSV	A*02:01	1.44	0.12	−0.59	1.56	0.97	3.9
HBVpol	455	GLSRYVARL	A*02:01	1.53	0.37	−2.08	1.90	−0.18	121
HDAg	26	KLEDLERDL	A*02:01	1.30	0.45	−3.68	1.75	−1.92	4734
HDAg	43	KLEDENPWL	A*02:01	1.54	0.45	−2.06	1.99	−0.07	114.3
HBVpol	173	SPYSWEQEL	B*35:01	1.54	0.38	−2.36	1.92	−0.44	229
HDAg	192	QGFPWDILF	B*35:01	1.27	1.10	−2.76	2.37	−0.38	571
HDAg	194	FPWDILFPA	B*35:01	0.66	−0.34	−1.20	0.32	−0.88	15.7

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
