# Peer review of "Adaptive Immune Responses, Immune Escape and Immune-Mediated Pathogenesis during HDV Infection"

_viruses, 2022, doi:10.3390/v14020198_

Round 1

Reviewer 1 Report

I want to congratulate the Authors for this authoritative, timely review, with extensive description of the relevant literature.  For sake of completeness, let me just ask to add and comment as needed the following papers:

1) Regarding the initial HDV RNA sequencing, please mention Kos A, et al. Nature 1986; 323:558-60

2) Regarding HDV pseudotypes (or lack thereof) in humans, please add and comment two very important papers, i.e., Cappy P, et al. J Infect Dis 2021;223:1376-80 and Roggenbach I, et al, Viruses 2021;13:1799

Author Response

We would like to thank the reviewer for the very kind general comment on our manuscript and also for the hint towards the three important additional references.

1) We have added the study by Kos et al. in line 36 of the revised manuscript (new reference #4).

2) We have added and discussed the two studies by Cappy et al. and Roggenbach et al. in lines 51-53 of the revised manuscript (new references #18 and 19).

Reviewer 2 Report

The review by Oberhardt V. et al. well depicts the current knowledge on innate and adaptive immune response against HDV infection, including mechanisms of viral escape. The review is well written and provides a comprehensive overview on this relevant topic.

I would suggest to the authors to shorten some parts that renders reading heavy as the last part of “CD8+ T cell response” paragraph (lines 275-296), particularly by maintaining only on the most relevant results of the papers by Landahl and Kefalakes.

Furthermore, the paragraph “viral escape” could be reorganized by focusing directly on HDV infection and, thus, by keeping just a short introduction on mechanisms of viral escape in the setting of other infections as HCV and HIV.

The authors should rephrase the sentence  at lines 427-432, since it is too long and difficult to follow.

The results from Landahl et al. should be better explained (lines 166-170)

Other minor comments:

line 54 precise should be changed in precisely

line 55 carrier should be carriers

line 69 check nucelos(t)ide

line 75 fro should be for

line 78 “)” is missing

line 144-145 Modify into “its related cellular and humoral immunity”

line 146 Initial should be changed into preliminary

line 157 With this strategy can be changed into By applying this strategy

line 201 Therefore and not therefor

line 223 “Of” is missing. The role of….

line 246 “, “ one comma should be removed

line 277 Overlapping is better than overlap

line 312 whereas is more proper than where

line 324 on the one side, remove “the”

line 331 it seems too strong to write “will not be recognized anymore”. I would suggest to soften the sentence by wrting recognized less efficiently

line 403 In contrast to e.g. can be changed into “in contrast to other viral infections”

Author Response

We would like to thank the reviewer for his/her very kind general comment on our manuscript as well as for the careful review. Please find our point-to-point response in the following:

1. I would suggest to the authors to shorten some parts that renders reading heavy as the last part of “CD8+ T cell response” paragraph (lines 275-296), particularly by maintaining only on the most relevant results of the papers by Landahl and Kefalakes.

=> Following the reviewer's suggestion, we have shortened the paragraph on the studies by Landahl and Kefalakes (lines 282-302 of the revised manuscript).

2. Furthermore, the paragraph “viral escape” could be reorganized by focusing directly on HDV infection and, thus, by keeping just a short introduction on mechanisms of viral escape in the setting of other infections as HCV and HIV.

=> The paragraph on viral escape has been shortened, directly focusing on viral escape in HDV infection (lines 328-344 of the revised manuscript). 

3. The authors should rephrase the sentence  at lines 427-432, since it is too long and difficult to follow.

=> The sentence has been rephrased (lines 434-440 of the revised manuscript).

4. The results from Landahl et al. should be better explained (lines 166-170).

=> The paragraph on the results by Landahl et al. has been better explained (lines 170-177 of the revised manuscript).

5. Line 54: precise should be changed in precisely

=> The error has been corrected (line 59 of the revised manuscript).

6. Line 55: carrier should be carriers

=> The error has been corrected (line 60 of the revised manuscript).

7. Line 69: check nucelos(t)ide

=> The error has been corrected (line 74 of the revised manuscript).

8. Line 75: fro should be for

=> The error has been corrected (line 80 of the revised manuscript).

9. Line 78 “)” is missing

=> The missing bracket has been added (line 83 of the revised manuscript).

10. Line 144-145: Modify into “its related cellular and humoral immunity”

=> The sentence has been modified (lines 149-150 of the revised manuscript).

11. Line 146: Initial should be changed into preliminary

=> "Initial" has been replaced by "preliminary" (line 151 of the revised manuscript).

12. Line 157: With this strategy can be changed into By applying this strategy

The sentence has been changed as suggested (line 162 of the revised manuscript).

13. Line 201: Therefore and not therefor

=> The error has been corrected (line 207 of the revised manuscript).

14. Line 223: “Of” is missing. The role of….

=> The missing word has been added (line 229 of the revised manuscript).

15. Line 246: “, “ one comma should be removed

=> The redundant comma has been removed (line 251 of the revised manuscript).

16. Line 277: Overlapping is better than overlap

=> "Overlap" has been changed to "overlapping" (line 284 of the revised manuscript).

17. Line 312: whereas is more proper than where

=> The error has been corrected (line 319 of the revised manuscript).

18. Line 324: on the one side, remove “the”

=> This sentence has been removed (see comment 3).

19. Line 331: it seems too strong to write “will not be recognized anymore”. I would suggest to soften the sentence by wrting recognized less efficiently

=> This sentence has been removed (see comment 3).

20. Line 403: In contrast to e.g. can be changed into “in contrast to other viral infections”

=> The sentence has been changed as suggested by the reviewer (line 411 of the revised manuscript).